# Through the Judge's Eyes: Inferred Thinking Traces Improve Reliability of LLM Raters

**Xingjian Zhang** *jimmyzxj@umich.edu*
*University of Michigan, Ann Arbor, Michigan, USA*

**Tianhong Gao** *tihgao@umich.edu*
*University of Michigan, Ann Arbor, Michigan, USA*

**Suliang Jin** *suliang@umich.edu*
*University of Michigan, Ann Arbor, Michigan, USA*

**Tianhao Wang** *tianhaowang@ucsd.edu*
*University of California, San Diego, California, USA*

**Teng Ye** *tengye@umn.edu*
*University of Minnesota, Twin Cities, Minneapolis, Minnesota, USA*

**Eytan Adar** *eadar@umich.edu*
*University of Michigan, Ann Arbor, Michigan, USA*

**Qiaozhu Mei** *qmei@umich.edu*
*University of Michigan, Ann Arbor, Michigan, USA*

**Reviewed on OpenReview:** *https://openreview.net/forum?id=1jLQ629Yps*

## Abstract

Large language models (LLMs) are increasingly used as raters for evaluation tasks. However, their reliability is often limited for subjective tasks, when human judgments involve subtle reasoning beyond annotation labels. **Thinking traces**, the reasoning behind a judgment, are highly informative but challenging to collect and curate. We present a human-LLM collaborative framework to infer thinking traces from label-only annotations. The proposed framework uses a simple and effective rejection sampling method to reconstruct these traces at scale. These inferred thinking traces are applied to two complementary tasks: (1) fine-tuning open LLM raters; and (2) synthesizing clearer annotation guidelines for proprietary LLM raters. Across multiple datasets, our methods lead to significantly improved LLM-human agreement. Additionally, the refined annotation guidelines increase agreement among different LLM models. These results suggest that LLMs can serve as practical proxies for otherwise unrevealed human thinking traces, enabling label-only corpora to be extended into thinking–trace–augmented resources that enhance the reliability of LLM raters.[1]

## 1 Introduction

Large language models (LLMs) are increasingly used as automated evaluators for open-ended text generation tasks, primarily due to their remarkable efficiency and scalability (Gu et al., 2024b; Zheng et al., 2023). This approach has been applied to evaluate diverse applications such as code generation (Zhou et al., 2025), text summarization (Bedemariam et al., 2025), and dialogue (Gu et al., 2024b). However, the reliability of LLM

---

[1]For reproducing results, our codebase can be accessed at: `https://github.com/xingjian-zhang/thru_judge_eye/tree/main`

Please read the prompt, the human story and the subject story (both stories might be the same). The story you will have to rate is the subject story.

Note: some stories have been abruptly cut in the middle of a sentence. Please rate them as if they ended just before the unfinished sentence.

Note: if the story is not relevant with respect to the prompt, it only affects the Relevance criterion! Do not rate 1 everywhere.

Then, please rate the subject story on a scale from 1 (worst) to 5 (best) on Complexity:

1: The story is very simple with no elaborate elements.
2: The story is somewhat simple with few elaborate elements.
3: The story has some complexity but lacks depth in certain areas.
4: The story is mostly complex with minor areas lacking depth.
5: The story is highly complex and elaborate throughout.

Figure 1: An example of the annotation codebook for evaluating complexity of short stories (Chhun et al., 2022). Only basic instructions and vague scoring rubrics are provided.

evaluators often decreases in subjective tasks that require nuanced human judgment (Ismayilzada et al., 2024; Gómez-Rodríguez & Williams, 2023), especially when they did not receive specialized training (Krumdick et al., 2025).

To understand what is missing in calibrating these models, we can draw an analogy from the process of training human annotators for qualitative coding tasks. To ensure alignment on a subjective task, a novice annotator typically benefits from three key components that an expert can provide: (1) an *annotation codebook* with detailed instructions and rubrics; (2) a set of *examples* illustrating representative inputs and their corresponding labels; and critically, (3) *explanations* detailing the reasoning for assigning a specific label to each example. Among the three components, explanations are particularly important in creating a shared understanding of the task among annotators (Artstein & Poesio, 2008; O'Connor & Joffe, 2020). We hypothesize that these explanations of the expert's reasoning, which we term the **thinking traces**, are equally necessary to calibrate LLM evaluators and improve their alignment with human judgment.

Despite their value, thinking traces are largely absent from existing annotation datasets. The primary reason is the substantial effort required: articulating and recording a detailed reasoning process is significantly more time-consuming and costly than providing a single label (DeYoung et al., 2019). Most annotation interfaces are not designed to capture this information, and incentivizing raters to produce high-quality traces is challenging (Carton et al., 2020). As a consequence, many datasets for subjective tasks are limited to sparse and sometimes unreliable labels, lacking the rich context that thinking traces would provide.

Reasoning language models (RLMs) offer a promising opportunity to address this issue (Comanici et al., 2025; Guo et al., 2025; Jaech et al., 2024). These models are designed to generate step-by-step reasoning, often termed "intermediate reasoning tokens", before producing a final answer (Guo et al., 2025). This model-generated reasoning is intended to align with the underlying thinking traces of human experts, also known as "human priors" in Guo et al. (2025). If this alignment is faithful, these models could serve as a proxy to reconstruct thinking traces at scale, mitigating the costly human annotation process. However, whether AI-generated reasoning truly aligns with human reasoning remains an open question, especially for subjective tasks where reasoning can be highly nuanced. While directly verifying this alignment is challenging, given the lack of ground-truth human thinking traces, a more pragmatic research question in our context is: **Can these model-inferred thinking traces, even if imperfect, serve as a useful proxy to improve the reliability of automated LLM raters?**

To tackle this question, we present a human-LLM collaborative framework to infer thinking traces from label-only human annotations. We employ a simple, yet effective, rejection sampling method to reconstruct these traces at scale. We demonstrate the utility of inferred thinking traces through two complementary applications. First, we use them as training data to fine-tune open-weight LLM raters so that they are specialized for the evaluation task. Second, for models where fine-tuning is not an option, we introduce a novel

method for codebook refinement. Inspired by iterative refinement practices in qualitative coding (O'Connor & Joffe, 2020), this approach addresses the common problem of ambiguous instructions in many annotation codebooks (Klie et al., 2023) (Figure 1). Our automatic pipeline leverages the inferred traces to synthesize a new, more explicit codebook that better steers the model's judgments. Experiments show that both methods significantly improve the alignment of LLM raters with human judgments. Furthermore, the synthesized codebook also improves the agreement between different LLM providers, demonstrating its utility in promoting cross-model consistency. These findings demonstrate that model-inferred thinking traces can effectively serve as a practical proxy for human reasoning, thus improving the reliability and consistency of automated LLM raters on subjective evaluation tasks.

## 2 Related Work

**Reasoning Language Models.** Reasoning Language Models (RLMs) represent a significant evolution from standard LLMs, as they are specifically trained to solve tasks that require multiple steps of deliberation. This class of models has become the de facto standard for high performance (Comanici et al., 2025; Guo et al., 2025; Jaech et al., 2024), demonstrating superior capabilities in tasks such as math and coding. Models such as DeepSeek R1 demonstrate the effectiveness of large-scale reinforcement learning in cultivating complex reasoning behaviors in these models (Guo et al., 2025), and provide public access to these reasoning tokens. Although these models are intended to align with "human priors" (Guo et al., 2025), the validity of these reasoning tokens remains a debatable question (Amirizaniani et al., 2024; Dasgupta et al., 2024; Mondorf & Plank, 2024). Testing the validity of thinking traces is beyond the scope of this paper. However, we are interested in a practical objective: leveraging RLMs as a tool to infer latent thinking traces from human experts, thereby improving the reliability of downstream LLM raters.

**Distilling Reasoning via Rejection Sampling.** A powerful paradigm for generating synthetic data is *rejection sampling fine-tuning*, where a model generates numerous candidate reasoning traces, and only those passing a filtering criterion are used for subsequent training. Existing work has pioneered this approach for tasks with objectively correct answers, such as mathematical reasoning, using an automatic verifier to accept only correct outputs (Guo et al., 2025; Zelikman et al., 2022; Wadhwa et al., 2024; Tong et al., 2024; Singh et al., 2023). To handle more nuanced tasks, methods such as RAFT (Dong et al., 2023) replaced the binary verifier with a learned reward model, filtering for outputs that score highly on desired attributes. This trend of applying reasoning to more diverse domains is also evident in recent work on recommender systems (Tsai et al., 2024), medical question-answering (Chen et al., 2024), software engineering (Cuadron et al., 2024; Penedo et al., 2025), finance (Qian et al., 2025), and cybersecurity (Yu et al., 2025). Concurrently, researchers have begun curating general-purpose SFT datasets enriched with distilled thinking traces to train open reasoning language models (Mattern et al., 2025; Guha et al., 2025). Crucially, while previous work exclusively focuses on using these curated traces for fine-tuning, we introduce a novel second application: distilling insights from the thinking traces to refine annotation codebooks automatically.

**Refining Codebooks for Qualitative Coding and Annotation** The refinement of codebooks is a long-standing challenge in both qualitative research and large-scale data annotation. Research in HCI and crowd-sourcing communities developed human-centric workflows to improve task instructions, often relying on crowd workers to identify ambiguities that were then resolved by an expert or through discussion (Chang et al., 2017; Manam & Quinn, 2018; K. Chaithanya Manam et al., 2019; Pradhan et al., 2022). More recently, researchers have begun to leverage LLMs to automate this process. This includes work on human-LLM collaboration for *qualitative coding* (Xiao et al., 2023; Halterman & Keith, 2025; Wiebe et al., 2025; Torii et al., 2024; Meng et al., 2024). Other work has demonstrated that LLMs can automatically analyze data or synthesize new guidelines to improve downstream performance (Bibal et al., 2025; Hsu et al., 2025; Srivastava et al., 2025). A key distinction of these automated methods is that they rely only on an LLM's intrinsic capabilities to analyze content or generate text, rather than being grounded in direct human annotation feedback. In contrast, our approach creates a collaborative pipeline with inputs from both humans and LLMs.

## 3 Method

### 3.1 Inferring Latent Thinking Traces

Our method begins with a seed dataset of human-annotated examples, denoted as $\mathcal{D}_{\text{human}} = \{(x_i, y_i)\}_{i=1}^{N}$. Here, $x_i$ represents an annotation target (e.g., a story) and $y_i$ is the corresponding label (e.g., a Likert rating for story 'complexity') assigned by human raters based on a specific codebook $\mathcal{C}$.

For any non-trivial annotation task, a human rater engages in a cognitive process to arrive at the final label $y_i$. We formalize this unobserved process as the human's **latent thinking trace**, $t_i$. This trace, which includes applying guidelines, resolving ambiguities, and weighing evidence, is not recorded in standard annotation workflows due to the high cost and inherent difficulty of articulating such complex reasoning.

Our goal is to generate a high-quality proxy for this latent trace, which we term the **reconstructed thinking trace**, $t_i'$. To achieve this, we use an RLM that can generate thinking tokens, hereafter referred to as the *generator model*, to perform rejection sampling. For each input item $x_i$, we provide the generator model with the codebook $\mathcal{C}$ and the annotation target and prompt it to generate a step-by-step thinking trace that concludes with a final rating. This generation is performed $k$ times independently, yielding a set of $k$ candidate trace-label pairs, $\{(t_{i,j}', y_{i,j}')\}_{j=1}^{k}$ where $t_{i,j}'$ denotes the $j$-th candidate trace and $y_{i,j}'$ denotes the $j$-th predicted label for $x_i$. For a model to reach the correct human label $y_i$, its reasoning is more likely (but not guaranteed) to be a plausible approximation of the latent human thought process. Therefore, we filter the candidate set by retaining only those traces where the model's predicted label $y_{i,j}'$ matches the ground-truth human label $y_i$. This creates a set of traces for each sample $x_i$:

$$T_i' = \{t_{i,j}', j \in \{1, \ldots, k\} \mid y_{i,j}' = y_i\}$$

From this set $T_i'$, we select the first sample to serve as the final reconstructed thinking trace, $t_i'$. This transforms our initial dataset into an augmented, reasoning-rich dataset, $\mathcal{D}_{\text{reason}} = \{(x_i, t_i', y_i)\}_{i=1}^{N}$, which forms the basis for the alignment techniques described next. In essence, each data point in $\mathcal{D}_{\text{reason}}$ can be considered a product of human-LLM collaboration, where the human label is used as a verifier to select an LLM-generated thinking trace. An example inferred thinking trace is shown in Figure 2 and an illustration of the overall process is provided in Figure 3.

---

Okay, let's tackle this rating. The user wants me to evaluate the Subject Story based on the Engagement criterion. The prompt mentions that stories create actual universes, and unfinished ones are hellish. The Human Story talks about a life controlled by a creator, leading to existential dread when stories are incomplete.

Now, looking at the Subject Story. It starts with a character exhausted from playing games, going home to feed his family. Then there's a sudden shift to a genie game over the ocean, hot water leaking, cold sea shattering the window. The narrative seems disjointed. The protagonist has a baseball bat, gets hit in the head, a storm hits, and the story ends abruptly.

Engagement-wise, the story is confusing. The transitions between scenes are jarring. The mix of mundane elements (family, baseball bat) with surreal events (genie, sudden storms) might intrigue some readers but could also lose them. The abrupt ending, while fitting the prompt's theme of unfinished stories, might leave readers frustrated. The imagery is vivid but lacks coherence, making it hard to follow. The emotional build-up from hope to despair isn't clearly developed here. The reader might struggle to connect with the protagonist or the plot. So, Engagement is probably low. I'd rate it a 2 because it has some interesting elements but is difficult to follow and loses interest quickly.

---

Figure 2: An example of the inferred thinking trace in evaluating the engagement level of a short story.

### 3.2 Improving LLM Raters

The primary goal of inferred thinking traces, $\mathcal{D}_{\text{reason}}$, is to improve the reliability of LLM raters. We show our method's utility in two complementary scenarios, which cover the primary ways LLMs are used in

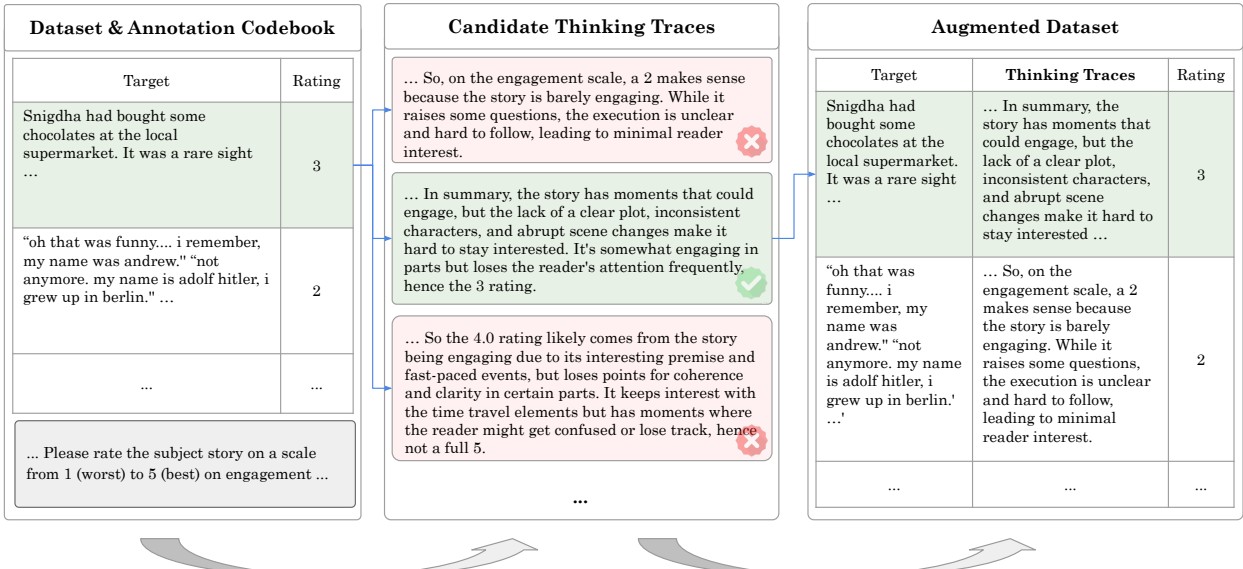

Figure 3: Illustration of Inferring Thinking Traces through an RLM. Multiple candidate thinking traces and labels are sampled. Only candidates that are aligned with human-annotated labels are preserved. The detailed process is provided in Section 3.1.

practice: as fine-tunable models that can be specialized to specific domains; and as black-box APIs. For open-weight models, we use $\mathcal{D}_{\text{reason}}$ to directly fine-tune the LLM rater, aligning its judgments with human reasoning patterns. More broadly, for the common case of proprietary models, we instead use our dataset to automatically refine the annotation codebook. This refined codebook serves as a more effective prompt to guide the model's behavior. This offers a practical, training-free way to leverage state-of-the-art LLMs.

**Fine-Tuning Specialized LLM Raters.** We show that the augmented dataset $\mathcal{D}_{\text{reason}}$ provides a powerful training resource for creating a specialized LLM rater. Unlike standard supervised fine-tuning (SFT), which only trains on the input-label pairs $(x_i, y_i)$, the augmented dataset provides the model with the complete reasoning trace $t_i'$. We hypothesize that the thinking traces contribute to a much richer training signal, which can significantly improve the model's alignment with human feedback. The model is trained using a standard SFT objective, as shown in Equation 1, to predict the full thinking trace followed by the final label. Here, $\pi_\theta$ denotes the LLM rater with weights $\theta$. This objective teaches the model not only *what* label to produce, but also *how* to reason towards it.

$$\mathcal{L}_{\text{SFT}} = \mathbb{E}_{(x, t', y) \sim \mathcal{D}_{\text{reason}}} \left[ -\log \pi_\theta(t', y|x) \right] \tag{1}$$

**Refining Annotation Codebooks.** We further investigate ways to take advantage of the inferred thinking traces when using proprietary LLMs whose weights are unavailable (i.e., they cannot be updated). Our approach is motivated by the observation that many codebooks contain ambiguous descriptions or lack concrete criteria, resulting in inconsistent ratings when used directly as prompts, as shown in Figure 1. Our collection of reconstructed traces serves as a corpus of successful step-by-step applications of these guidelines. We can leverage these traces to improve the codebook by extracting common reasoning patterns and synthesizing a more explicit step-by-step procedure. This improved codebook, $\mathcal{C}'$, can then be used to construct more effective prompts for proprietary LLMs where fine-tuning is not an option.[2]

Specifically, we propose a two-stage process that targets the two main components of the codebook: the *task instructions* and the *scoring rubric*.

---

[2]While we do not test this hypothesis in our work, an improved codebook could also potentially enhance the consistency and quality of human raters.

1. *Improve task instructions*: We first sample 10 thinking traces for each rating level and prompt an LLM to summarize the common reasoning patterns in these traces into an explicit, step-by-step procedure for the new task instructions.

2. *Enrich scoring rubrics:* To enrich the scoring rubric, we then sample a set of 50 thinking traces for each rating level and extract short critiques from each one (e.g., "The lack of a coherent plot..."). Not all critiques are representative of a given rating level. For example, a good story may still have some flaws. To find the most representative critiques, we cluster the text embeddings of these critiques using the `text-embedding-3-large` model and select the critiques with the highest semantic similarity to each cluster's centroid. Finally, we present these representative critiques to an LLM, prompting it to synthesize a new, detailed rubric for each rating level that is grounded in concrete examples.

We provide the complete prompts for this process in Appendix D.

---

**Read the Materials Thoroughly**  Start by reading the prompt, the human story, and the subject story. Note that the subject story is the one being rated, not the human story . . .

**Step-by-Step Rating**
- Look for key elements such as characters, events, plot, themes, or setting . . .
- Evaluate if the identified elements are developed, interconnected, and coherent . . .
- Consider whether the story follows a structured progression (linear or non-linear) . . .
- Stories with minimal elements or undeveloped content are simpler. Stories that attempt multi-layered exploration of ideas, even if not fully realized, should be evaluated as more complex.

**Select the Rating**

1. The story is very simple, lacks depth, and introduces no intricate or developed elements.

2. The story is somewhat simple, with few elements that are underdeveloped, fragmented, or disconnected.

3. The story demonstrates some complexity, introducing multiple elements, but lacks depth, development, or coherence in integrating these elements.

4. The story is mostly complex, weaving together several developed elements with minor areas that could be expanded or refined.

5. The story is highly elaborate and complex, integrating multiple layers (e.g., detailed world-building, character depth, advanced structure, thematic sophistication) in a cohesive and interconnected way.

**Write the final rating**  Place your chosen rating . . .

---

Figure 4: An example of the refined annotation codebook. Some content has been omitted due to space limitations. See Appendix E for a complete example.

## 3.3  Evaluation

To empirically validate LLM raters, we use two primary criteria. The first, *Agreement with Human Judgments*, serves as the main test for both applications. The second, *Inter-Rater Reliability*, is an assessment designed to measure the specific impact of our codebook refinement. All evaluations are performed on held-out test sets randomly sampled from the original dataset.

**Agreement with Human Judgments**  This criterion measures the degree to which an LLM rater's outputs agree with human judgments. We apply this evaluation to the outputs of both our applications. For this analysis, we treat a held-out set of human ratings as the gold standard and compare the LLM-generated labels against them. To quantify this agreement, we use several standard metrics for labels in the Likert

scale. Our primary focus is on **Kendall's Tau ($\tau$)**, as ranking correlation[3] is the most used measure in the literature (Liu et al., 2023; 2024; Thakur et al., 2025; Gu et al., 2024a). For a more complete assessment, we also report the Mean Squared Error (MSE) for average error magnitude and the Intraclass Correlation Coefficient (ICC3) for consistency, based on a two-way mixed-effects model (Koo & Li, 2016).

**Inter-Rater Reliability.** For codebook refinement, another robust indicator of success is whether $\mathcal{C}'$ enables independent raters to arrive at the same conclusions consistently. High inter-rater reliability is a sign of a clear and well-defined annotation process. To measure this, we use the improved codebook $\mathcal{C}'$ to prompt $M$ different proprietary LLMs to rate the same set of items. We then calculate the **reliability across these $M$ independent LLM raters** using ICC3 (Koo & Li, 2016).

**Statistical Significance Test.** To assess statistical significance, we use the following statistical tests. For fine-tuned LLM raters, we apply a paired $t$-test on MSE, and employ one-sided bootstrap hypothesis tests to evaluate whether improvements in $\tau$ and ICC3 are statistically significant in the direction of superiority. For the refined codebook analysis, we conduct one-sided paired $t$-tests on each metric, comparing the performance of four LLM raters before and after refinement, thereby directly testing whether the refinement leads to consistent improvements across raters. To assess the statistical significance of agreement between LLM raters, we conduct a one-sided bootstrap hypothesis test to evaluate whether the improvement in ICC3 is significant in the direction of superiority.

## 4 Experiment

### 4.1 Datasets

We evaluated our framework across five diverse annotation tasks, all of which require deliberate reasoning rather than simple intuition to assign a reliable score. Another key selection criterion is the public availability of the annotation codebook used for each task. These tasks cover two distinct types of human feedback on generated text: single holistic quality scores and fine-grained scores across multiple dimensions. For consistency in our evaluation methodology, we focused on datasets that use a Likert scale. However, our framework is readily applicable to other formats, such as forced-choice rankings. The five tasks are summarized below:

1. **Evaluating Short Stories.** We use the HANNA dataset (Chhun et al., 2022), an annotation dataset for evaluating machine-generated stories. The stories were rated by human annotators recruited through Amazon Mechanical Turk, restricted to native English speakers with a Master's Qualification. The *complexity* dimension provides a measure of structural and elemental richness, while *engagement* captures readers' emotional involvement, reflecting a more subjective perception.

2. **Evaluating Student-Written Essays.** For this task, we utilize a dataset of student essays from a Kaggle competition on automated essay scoring (Crossley et al., 2024). Each essay is assigned a single *holistic* score by expert human graders.

3. **Evaluating News Summaries.** We use the SummEval benchmark (Fabbri et al., 2020), which provides human evaluations of machine-generated summaries of news articles. Each summary is annotated by three expert annotators with prior experience in writing research paper summaries for academic conferences. We focus on the *consistency* and *fluency*, which respectively evaluate factual correctness and linguistic quality, two key aspects of summary reliability and readability.

4. **Evaluating Translations.** The WMT 2020 dataset (Freitag et al., 2021) provides expert human evaluations of machine translation outputs under the Scalar Quality Metric framework. We focus on the Chinese-to-English translation task, where each translation is given a *holistic* quality score by three professional translators native in the target language.

5. **Evaluating Chatbot Responses.** We use the HelpSteer2 dataset (Wang et al., 2024), which contains human feedback on AI-generated conversational responses. The annotations were collected

---

[3]We observe a high correlation between Spearman's $\rho$ and Kendall's $\tau$ in the results. We only report $\tau$ here for simplicity.

from trained human raters on the Scale AI platform, where each response was evaluated by at least three annotators across five quality dimensions. Additional annotators were assigned when initial ratings showed high disagreement to ensure reliability. We focus on *helpfulness*, which measures informativeness and relevance of the response, and *correctness*, which evaluates the factual accuracy.

More statistics and pre-processing details for each dataset are provided in Appendix B.

## 4.2 Data Processing

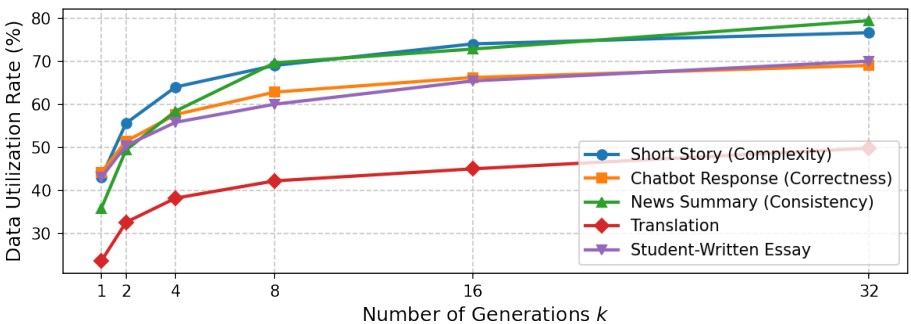

Figure 5: Trade-off between Data Utilization Rate and Number of Samples. Data utilization rate denotes the percentage of data points that have matched thinking traces within the $k$ generations.

**Filter Gold Standards** The evaluation of an LLM rater requires comparing its judgments with a reliable gold standard derived from human consensus. For datasets that already provide a single aggregated human score per sample, such as Essay and HelpSteer2, we adopt this value directly as our gold standard. For datasets that provide raw ratings from multiple annotators, such as HANNA, SummEval, and WMT-2020, we construct the gold standard by first filtering for quality. We discard contentious examples where the standard deviation of human scores exceeds 1.0, and then define the gold standard as the median of the remaining ratings.

**Infer Thinking Traces** As outlined in Section 3, we use rejection sampling to infer thinking traces from the annotation datasets. The original annotation codebook from each source serves directly as the prompt. For our generator models, we selected `DeepSeek-R1` and `gpt-oss-120b` as they output complete thinking traces. We prompt each model to generate sixteen ($k = 16$) thinking traces for every sample and keep the first trace that aligns with the human annotator's final rating. Here, the selection of $k$ is a pragmatic trade-off between computational budget and the coverage of matched valid thinking traces, as shown in Figure 5. A sampling temperature of 1.0 is used during inference to encourage generative diversity.

## 4.3 Fine-Tuning Specialized LLM Rater

This experiment demonstrates how inferred thinking traces can be leveraged to train more effective automated evaluators via reasoning-enhanced supervised fine-tuning.

We use `DeepSeek-R1-0528-Qwen3-8B` and `DeepSeek-R1-Distill-Llama-8B` as our base model (Guo et al., 2025). The model is then fine-tuned on each individual task separately. During fine-tuning, the model is trained not only to predict the final rating but also to generate the entire thought process. This approach encourages the model to learn the underlying reasoning process that leads to a specific judgment. To prevent overfitting, we use early stopping with a patience of 100 steps on a held-out validation set, selecting the model checkpoint that achieves the highest Kendall's $\tau$ correlation with human ratings. The performance of this final model is then evaluated by comparing its ratings against the human gold standard on the test set.

| Task | Original Model | | | SFT (DeepSeek Trace) | | | SFT (gpt-oss Trace) | | |
|------|------|------|------|------|------|------|------|------|------|
| Metric | $\tau$ | ICC$_3$ | MSE | $\tau$ | ICC$_3$ | MSE | $\tau$ | ICC$_3$ | MSE |
| Short Story (Complexity) | 0.177 | 0.231 | 2.016 | 0.463* | 0.604* | 0.838* | 0.460* | 0.581* | 0.944* |
| Short Story (Engagement) | 0.275 | 0.289 | 1.939 | 0.048 | 0.364 | 1.364 | 0.179 | 0.371 | 1.831 |
| Student-Written Essay | 0.202 | 0.233 | 1.947 | 0.322* | 0.267 | 1.149* | 0.240 | 0.244 | 1.387* |
| News Summary (Consistency) | 0.141 | 0.186 | 2.904 | 0.276 | 0.319 | 3.476 | 0.243 | 0.331 | 2.970 |
| News Summary (Fluency) | 0.275 | 0.378 | 1.938 | 0.319 | 0.402 | 1.160* | 0.306 | 0.498 | 1.200* |
| Translation | 0.171 | 0.240 | 2.778 | 0.260* | 0.348* | 2.492 | 0.256* | 0.314* | 2.439* |
| Chatbot Response (Correctness) | 0.162 | 0.288 | 2.049 | 0.297* | 0.403* | 1.815* | 0.263* | 0.433* | 1.897 |
| Chatbot Response (Helpfulness) | 0.176 | 0.274 | 1.969 | 0.264* | 0.422* | 1.723* | 0.295* | 0.466* | 1.575* |
| **Average** | **0.197** | **0.265** | **2.193** | **0.281** | **0.391** | **1.752** | **0.280** | **0.405** | **1.780** |

Table 1: Effect of Reasoning-Enhanced SFT on Human-LLM Rater Agreement (`DeepSeek-R1-Distill-Qwen3-8B`). For the metrics, higher is better for Kendall's $\tau$ and ICC$_3$, while lower is better for MSE. An asterisk (*) denotes a statistically significant improvement over the baseline ($p < 0.05$).

**Results and Analysis**  The results in Table 1 show that reasoning-enhanced SFT significantly improves the performance of the LLM rater[4]. Across all tasks, the models fine-tuned on inferred traces from both `DeepSeek-R1` and `gpt-oss-120b` achieve significant gains over the original model. On average, reasoning-enhanced SFT improves Kendall's $\tau$ from 0.197 to 0.281, a relative increase of 42.6%. Notably, the final performance of models trained on traces from `DeepSeek-R1` and `gpt-oss-120b` is highly comparable. Both sets of inferred traces, despite originating from different models, serve as effective training data to improve the base rater. This demonstrates the universality of our framework over heterogeneous RLMs.

An outlier is the *Short Story (Engagement)* task, where both SFT models show a degradation in performance, particularly in Kendall's $\tau$. We hypothesize that this anomaly may stem from a potential lack of sufficient high-quality data points in the seed dataset for this specific dimension, making it difficult to learn a consistent reasoning pattern.

Overall, despite some task-specific variations, the evidence supports that **fine-tuning on inferred thinking traces is an effective method for enhancing the alignment of LLM raters with human judgments**.

## 4.4   Refining Annotation Codebook

| Task | Original Codebook | | | Refined (DeepSeek Trace) | | | Refined (gpt-oss Trace) | | |
|------|------|------|------|------|------|------|------|------|------|
| Metric | $\tau$ | ICC$_3$ | MSE | $\tau$ | ICC$_3$ | MSE | $\tau$ | ICC$_3$ | MSE |
| Short Story (Complexity) | 0.428 | 0.497 | 1.116 | 0.512 | 0.571 | 0.634* | 0.401 | 0.504 | 0.999 |
| Short Story (Engagement) | 0.303 | 0.514 | 2.044 | 0.380 | 0.559 | 1.525 | 0.391* | 0.564 | 1.392 |
| Student-Written Essay | 0.421 | 0.413 | 0.945 | 0.491* | 0.481 | 0.895 | 0.459* | 0.433 | 0.957 |
| News Summary (Consistency) | 0.166 | 0.150 | 2.861 | 0.188 | 0.178 | 2.591 | 0.177 | 0.181 | 2.668 |
| News Summary (Fluency) | 0.142 | 0.182 | 3.121 | 0.192 | 0.197 | 2.203 | 0.190 | 0.248 | 2.855 |
| Translation | 0.347 | 0.419 | 2.392 | 0.368 | 0.448* | 1.942* | 0.356 | 0.452 | 1.953* |
| Chatbot Response (Correctness) | 0.354 | 0.451 | 1.920 | 0.370* | 0.485 | 1.724* | 0.378* | 0.484 | 1.804 |
| Chatbot Response (Helpfulness) | 0.366 | 0.475 | 1.801 | 0.390* | 0.511* | 1.642* | 0.376 | 0.484 | 1.585* |
| **Average** | **0.316** | **0.388** | **2.025** | **0.361** | **0.429** | **1.645** | **0.341** | **0.419** | **1.777** |

Table 2: Effect of Codebook Refinement on Human-LLM Rater Agreement. Each value in the table is an average of the corresponding metrics calculated for each LLM rater.

Although SFT is effective, it is limited to LLMs for which weights are available to update. In more common cases, we have access to the state-of-the-art model through black-box APIs only. To steer their behavior, we need to synthesize more effective prompts in the same way that we refine the annotation codebook for human raters. Following the procedure described in Sec 3, we generate two refined annotation codebooks for each task using traces from `DeepSeek-R1` and `gpt-oss-120b`, respectively. We test the effective-

---

[4]Due to space limit, we put the results for `DeepSeek-R1-Distill-Llama-8B` in Appendix A.

| Task | Original Codebook | Refined (DeepSeek Trace) | Refined (gpt-oss Trace) |
|------|-------------------|--------------------------|--------------------------|
| Short Story (Complexity) | 0.613 | 0.700 | 0.771* |
| Short Story (Engagement) | 0.883 | 0.810 | 0.823 |
| Student-Written Essay | 0.560 | 0.602* | 0.554 |
| News Summary (Consistency) | 0.405 | 0.501* | 0.417 |
| News Summary (Fluency) | 0.352 | 0.395 | 0.370 |
| Translation | 0.633 | 0.672* | 0.689* |
| Chatbot Response (Correctness) | 0.578 | 0.649* | 0.628* |
| Chatbot Response (Helpfulness) | 0.615 | 0.636* | 0.604 |
| **Average** | **0.580** | **0.621** | **0.607** |

Table 3: Effect of Codebook Refinement on Rater Agreement among LLM Raters ($\text{ICC}_3$).

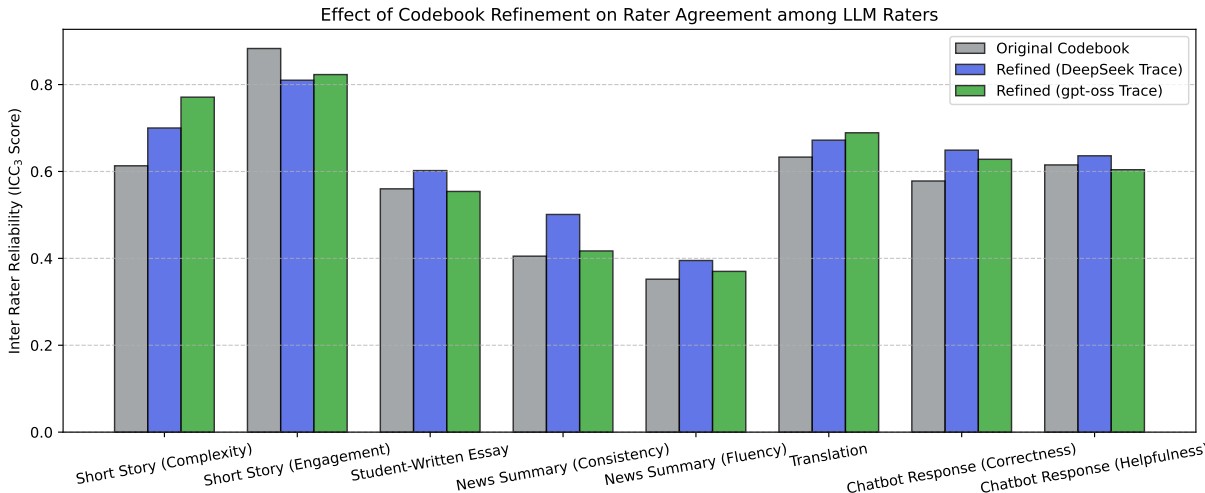

Figure 6: Effect of Codebook Refinement on Rater Agreement among LLM Raters ($\text{ICC}_3$).

ness of these refined codebooks on four powerful LLMs from different model providers: `Claude-4-Sonnet`, `Gemini-2.5-Flash`, `GPT-5`, and `DeepSeek V3.1`. We adopt a temperature of 1.0 and sampling probability mass cutoff of 0.95 for inference.

**Improved Agreement with Human Judgments.** Table 2 shows the **improved average agreement between four LLM raters and human judgments**. On average, the codebooks refined by `DeepSeek-R1` traces improve human-LLM Kendall's $\tau$ from 0.316 to 0.361 (a 14.2% improvement), and the codebooks refined by `gpt-oss-120b` traces improve $\tau$ to 0.341 (a 7.9% improvement), showing a consistent improvement. Additionally, the ratings obtained by prompting more advanced LLMs with the refined codebook outperform those by fine-tuned smaller LLM raters (in Table 1). This highlights the effectiveness and practicality of refining the annotation codebook.

**Improved Inter-Rater Reliability.** Crucially, **the refined codebooks also improve inter-rater reliability among different LLM raters**. Table 3 shows that the average inter-rater agreement ($\text{ICC}_3$) increases from 0.580 to 0.621 with the thinking traces inferred by `DeepSeek-R1`. The benefits are most pronounced on tasks with relative objective criteria, such as correctness of *Chatbot Response* and *Translation*. Conversely, on highly subjective tasks like evaluating the engagement of *Short Story*, the refined codebooks can sometimes increase disagreement between LLM raters. We hypothesize that this decrease is not necessarily a negative indicator: the original codebook may have induced high but superficial agreement by promoting a shared systematic bias among the LLM raters. By providing more nuanced guidance, the refined codebook might be reducing this shared bias, potentially leading the models to better reflect the natural and sometimes divergent distribution of human judgments.

**Claude (Before):** . . . The subject story is unrelated to the prompt. It mostly describes a mundane routine with some disjointed details. . . Then shifts abruptly to snow and vague dialogue about a "gift" and "danger," but these are underdeveloped. . . . The narrative lacks structure, coherence, or meaningful plot development. . . . Elements that could add depth are too vague. . . . The story reads like disconnected fragments with no complexity. <rating>1</rating>
**Claude (After):** . . . The story includes some elements: narrator routine, home setting, dialogue, and atmosphere. . . . But development is weak: abrupt jumps, unexplained characters, disconnected dialogue, no plot or character growth, abrupt ending. . . . Though it introduces multiple ideas (routine, relationships, mysterious "gift"), they are fragmented and underdeveloped. . . . Falls into the category of "few elements, underdeveloped, fragmented, disconnected." <rating>2</rating>
**DeepSeek (Before):** . . . The subject story is different from the prompt, lacks magical elements. . . . Contains surreal and disjointed aspects (time reversing, "gift," ambiguous dialogue) that add some complexity. . . . But overall lacks cohesion and depth, feels fragmented. <rating>3</rating>
**DeepSeek (After):** . . . Identifies elements: narrator, teacher, child; mundane events; home/snow setting; vague themes ("gift," "danger"). . . . Notes minimal depth, low coherence, abrupt time shifts, unresolved dialogue. . . . Structure linear but disjointed. . . . Introduces some elements but underdeveloped, no advanced techniques. . . . Story is somewhat simple with disconnected elements. <rating>2</rating>

Figure 7: Example thinking traces from two LLM raters on evaluating the complexity of a short story, before and after codebook refinement.

**Qualitative Analysis of Annotation before and after Refinement.** The example in Figure 7 provides a clear illustration of how the refined codebook improves rater alignment. Initially, using the original codebook to evaluate a story (ground truth rating: 2), the two LLM raters produce divergent and incorrect scores. This disagreement stems from two core flaws in the original guide.

First, it lacks a **step-by-step procedure**, leading to inconsistent evaluation paths. Claude, for example, immediately makes a top-down judgment about the story's "lack of structure, coherence, or meaningful plot development." In contrast, DeepSeek takes a bottom-up approach, first identifying "surreal and disjointed aspects" which it views as a source of "complexity." The refined codebook solves this by introducing a clear, standardized process. As seen in the new traces, both raters now begin by systematically identifying the story's elements—such as "narrator routine, home setting, dialogue" (Claude) and "narrator, teacher, child; mundane events" (DeepSeek)—before making a final judgment.

Second, the original **scoring rubric is too abstract**, forcing the models to weigh competing factors differently. The refined rubric corrects this by grounding its descriptions in **concrete examples**. This allows both raters to converge on the same nuanced conclusion. Claude's final justification now states that the story "falls into the category of 'few elements, underdeveloped, fragmented, disconnected,'" while DeepSeek similarly concludes it is "somewhat simple with disconnected elements." They both correctly assign a rating of 2, demonstrating that the refined codebook aligns LLM raters by standardizing the entire evaluation process.

### 4.5 Ablation Study

**Ablation on Thinking Traces in SFT** To disentangle the contribution of reasoning traces from the general benefits of SFT, we trained the same base model using only the final ratings from all available training data across all tasks. Table 4 shows that the model trained exclusively on ratings suffered a performance drop across multiple tasks compared to the trace-augmented model, resulting in overall performance slightly worse than the baseline. This suggests that the model's inability to utilize intermediate reasoning steps during inference hinders its evaluation capability. Therefore, these results confirm that the generated thinking traces are an essential component for grounding the model's final judgments.

**Ablation on Refinement Components** To understand the individual contributions of our two-stage refinement process, we conduct an ablation study using thinking traces from `DeepSeek R1` on three diverse

| Task | Original Model | | | SFT w/ traces | | | SFT w/o traces | | |
|---|---|---|---|---|---|---|---|---|---|
| Metric | $\tau$ | $ICC_3$ | MSE | $\tau$ | $ICC_3$ | MSE | $\tau$ | $ICC_3$ | MSE |
| Short Story (Complexity) | 0.177 | 0.231 | 2.016 | 0.463* | 0.604* | 0.838* | 0.329 | 0.312 | 1.478 |
| Short Story (Engagement) | 0.275 | 0.289 | 1.939 | 0.048 | 0.364 | 1.364 | 0.166 | 0.182 | 3.101 |
| Student-Written Essay | 0.202 | 0.233 | 1.947 | 0.322* | 0.267 | 1.149* | 0.283* | 0.277* | 2.036 |
| News Summary (Consistency) | 0.141 | 0.186 | 2.904 | 0.276 | 0.319 | 3.476 | 0.193 | 0.236 | 2.666 |
| News Summary (Fluency) | 0.275 | 0.378 | 1.938 | 0.319 | 0.402 | 1.160* | 0.167 | 0.381 | 1.565 |
| Translation | 0.171 | 0.240 | 2.778 | 0.260* | 0.348* | 2.492 | 0.217 | 0.272 | 2.763 |
| Chatbot Response (Correctness) | 0.162 | 0.288 | 2.049 | 0.297* | 0.403* | 1.815* | 0.103 | 0.140 | 2.742 |
| Chatbot Response (Helpfulness) | 0.176 | 0.274 | 1.969 | 0.264* | 0.422* | 1.723* | 0.072 | 0.120 | 2.430 |
| **Average** | **0.197** | **0.265** | **2.193** | **0.281** | **0.391** | **1.752** | **0.191** | **0.240** | **2.348** |

Table 4: Ablation Study on the Effect of Thinking Traces in SFT.

| Task | Original | | | Refined (Instruction) | | | Refined (Rubric) | | | Refined (Both) | | |
|---|---|---|---|---|---|---|---|---|---|---|---|---|
| Metric | $\tau$ | $ICC_3$ | MSE | $\tau$ | $ICC_3$ | MSE | $\tau$ | $ICC_3$ | MSE | $\tau$ | $ICC_3$ | MSE |
| Story | 0.303 | 0.514 | 2.044 | 0.372 | 0.551 | 1.867 | 0.380 | 0.537 | 1.701 | 0.380 | 0.559 | 1.525 |
| Chatbot | 0.354 | 0.451 | 1.920 | 0.374 | 0.480 | 1.827 | 0.359 | 0.453 | 1.836 | 0.370 | 0.485 | 1.724 |
| Summary | 0.166 | 0.150 | 2.861 | 0.168 | 0.145 | 2.405 | 0.185 | 0.195 | 2.602 | 0.188 | 0.178 | 2.591 |

Table 5: Ablation study on the effect of refinement codebook component. Story: Short Story (Engagement). Chatbot: Chatbot Response (Correctness). Summary: News Summary (Consistency).

tasks: *Short Story (Engagement)*, *Chatbot Response (Correctness)*, and *News Summary (Consistency)*. To isolate the effect of each component, we create two separate versions of each codebook: one with only the task instructions refined and another with only the scoring rubric refined, and then assess Human-LLM rater alignment. Results in Table 5 show that **both instruction refinement and rubric refinement enhance performance, though their relative impact depends on the specific weaknesses of the original codebook**. For the *Chatbot Response* task, instruction refinement yields larger gains, increasing Kendall's $\tau$ from 0.354 to 0.374, compared to 0.359 from rubric refinement. In contrast, rubric refinement proves more effective for *News Summary*, raising $\tau$ from 0.166 to 0.185, whereas instruction refinement provides only a marginal increase to 0.168. For the *Short Story* task, both refinements deliver comparably strong improvements. This analysis reveals that the optimal refinement strategy depends on the original codebook's deficiencies: rubric refinement is most beneficial when the original lacks a detailed rating scale (as with News Summary), while instruction refinement is critical when it lacks a step-by-step guideline, even with a detailed rubric (as with Chatbot Response). These findings validate our two-stage approach, highlighting its robustness in addressing distinct types of flaws in annotation guidelines.

| Task | Original Codebook | | | Refined (Thinking Tokens) | | | Refined (Post Hoc) | | |
|---|---|---|---|---|---|---|---|---|---|
| Metric | $\tau$ | $ICC_3$ | MSE | $\tau$ | $ICC_3$ | MSE | $\tau$ | $ICC_3$ | MSE |
| Story | 0.303 | 0.514 | 2.044 | 0.380 | 0.559 | 1.525 | 0.330 | 0.533 | 1.743 |
| Chatbot | 0.354 | 0.451 | 1.920 | 0.370 | 0.485 | 1.724 | 0.376 | 0.484 | 2.275 |
| Summary | 0.166 | 0.150 | 2.861 | 0.188 | 0.178 | 2.591 | 0.172 | 0.160 | 2.533 |

Table 6: Ablation Study on using post hoc explanation.

**Ablation on Using Post Hoc Explanations.** A key prerequisite to reconstructing effective thinking traces is full access to the RLM thinking tokens. However, this may not always be possible, especially for state-of-the-art model like `gpt-5` and `Gemini-2.5-Pro`, which only expose a brief summary of their internal thinking. Therefore, we investigate whether our framework can be adapted for models that do not expose their internal thinking process, such as most proprietary LLMs. To address this, we test the usage of *post hoc explanations* as an alternative for pre-decision thinking traces. Data collection involves providing a generator model with the annotation target $x_i$ and the gold standard human rating $y_i$, and prompting it to generate multiple post hoc candidate explanations for how one could arrive at that rating. These candidates are

then filtered for quality: we remove all explicit rating information from the generated text and use another LLM to verify if the remaining reasoning still leads to the correct rating $y_i$. Due to multiple rounds of LLM calls, this method is more costly. The results in Table 6 show that this collection pipeline remains a highly effective approach. The codebook refined using post hoc explanations significantly outperforms the original codebook on both tasks. While its performance is slightly lower than using native thinking traces for the Story task, it is highly comparable for the News Summary (consistency) task. This finding demonstrates the robustness of our framework, suggesting that it can be successfully extended to LLMs without explicit thinking tokens.

## 5 Discussion

We list several limitations and potential directions that could be addressed in future research:

**Sampling Strategy**   Our rejection sampling approach is effective for inferring thinking traces when the generator model is capable enough to overlap with human judgment. However, its efficacy may be reduced in tasks with large alignment gaps. The model might struggle to generate a trace that matches the human's final label, even when multiple candidate responses are sampled. To mitigate this, future research could explore sampling strategies that enable a broader coverage of possible thinking traces in the space. In addition, our current method infers thinking traces for an aggregated rating from multiple raters. Future work could instead infer thinking traces for each individual annotator, which may enrich the diversity of collected reasoning patterns and provide deeper insights into subjective variation across raters.

**Relation with Automated Prompt Optimization**   While beyond the scope of this paper, our codebook refinement process can also be viewed as a complementary approach to Automated Prompt Optimization (APO) techniques (Pryzant et al., 2023). Many APO methods iteratively refine the prompts, starting from a simple seed that is hand-crafted (Ramnath et al., 2025). Our method, in contrast, provides a semantically rich and data-driven starting point derived from inferred human thinking traces. This refined codebook could serve as a high-quality initial prompt, which can then be passed to established APO algorithms for further fine-tuning. This two-stage approach could bridge the gap between human-centric guideline design and automated optimization, leading to prompts that are both effective and grounded in human cognitive patterns.

**Involving Human Annotators**   Another promising direction is to create a lightweight form of human-AI collaboration for future data annotation. This approach achieves a practical tradeoff between the high complexity of asking humans to write thinking traces from scratch and the need for an accurate thinking trace. Future workflows could first use a model to generate several candidate thinking traces given a human rating. A human rater would then simply select the trace that best reflects their own reasoning process. This "select-and-validate" method is significantly less demanding for annotators, but still yields the high-fidelity data needed to further strengthen the reliability of downstream LLM raters. Furthermore, inferred thinking traces have immediate applications for human-centered tasks. They can serve as valuable training resources to accelerate the onboarding of new human annotators, and the refined annotation codebooks could be used not only to guide LLMs but also to improve inter-rater reliability among human teams – a hypothesis that we can test in future work.

## 6 Conclusion

Our work introduces a scalable framework to infer the latent thinking traces of human annotators using reasoning language models. We demonstrate that these inferred traces serve as high-fidelity proxies for the reasoning paths of human judges across two complementary case studies. First, their integration into reasoning-enhanced fine-tuning significantly improves the alignment of open-weight LLM raters to human ratings. Second, these traces can be used to automatically refine annotation guidelines, which in turn boosts agreement among multiple LLM raters, as well as their agreement with human judgments. Our findings

highlight the potential to unlock vast, label-only datasets, transforming them into transparent, reasoning-rich resources for building more reliable and interpretable LLM-based evaluators.

**Acknowledgments**

This work was in part supported by a Propelling Original Data Science (PODS) grant sponsored by Michigan Institute for Data & AI in Society (MIDAS) and Microsoft.

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

# A    Additional Tables

**Generalizability across Base Architectures**    To demonstrate that our framework is not limited to specific model families, we conducted experiments using DeepSeek-R1-Distill-Llama-8B as the base model. Table 7 reports the Human-LLM agreement metrics before and after applying our reasoning-enhanced SFT. The results show consistent improvements across the majority of tasks, confirming that our method effectively generalizes to Llama-based architectures.

**Sensitivity to Trace Quantity**    We further investigated the robustness of our codebook refinement stage regarding the number of thinking traces used. Table 8 presents an ablation study on the *Short Story (Engagement)* task, varying the number of traces for instruction optimization ($N_{inst}$) and rubric generation ($N_{rubric}$). We observe that performance improves as more traces are utilized, plateauing around our selected configuration (10/50). Crucially, even minimal trace settings (5/20) outperform the original codebook, indicating that the method is data-efficient and not overly sensitive to hyperparameter tuning.

| Task | Original Model | | | SFT (DeepSeek Trace) | | |
|------|:---:|:---:|:---:|:---:|:---:|:---:|
| Metric | $\tau$ | ICC$_3$ | MSE | $\tau$ | ICC$_3$ | MSE |
| Short Story (Complexity) | 0.324 | 0.419 | 1.552 | 0.315 | 0.419 | 1.248 |
| Short Story (Engagement) | 0.305 | 0.312 | 1.199 | 0.292 | 0.336 | 1.108 |
| Student-Written Essay | 0.075 | 0.087 | 4.475 | 0.237* | 0.260* | 1.553* |
| News Summary (Consistency) | 0.059 | 0.0418 | 1.782 | 0.128 | 0.146 | 4.018 |
| News Summary (Fluency) | 0.159 | 0.203 | 1.792 | 0.213 | 0.359 | 1.203* |
| Translation | 0.174 | 0.143 | 2.145 | 0.236* | 0.173 | 2.028* |
| Chatbot Response (Correctness) | 0.063 | 0.072 | 2.319 | 0.154* | 0.200* | 2.046* |
| Chatbot Response (Helpfulness) | 0.053 | 0.070 | 2.434 | 0.155* | 0.164* | 2.228* |
| **Average** | **0.151** | **0.168** | **2.212** | **0.216** | **0.257** | **1.929** |

Table 7: Effect of Reasoning-Enhanced SFT on Human-LLM Rater Agreement (`DeepSeek-R1-Distill-Llama-8B`). For the metrics, higher is better for ICC$_3$ and Kendall's $\tau$, while lower is better for MSE. An asterisk (*) denotes a statistically significant improvement over the baseline ($p < 0.05$).

| Setup ($N_{inst}$ / $N_{rubric}$) | $\tau$ | ICC | MSE |
|------|:---:|:---:|:---:|
| Original Codebook | 0.303 | 0.514 | 2.044 |
| 5 / 20 | 0.370 | 0.548 | 1.622 |
| 10 / 50 *(Selected)* | 0.380 | **0.559** | **1.525** |
| 20 / 60 | **0.396** | 0.549 | 1.737 |
| 30 / 70 | 0.363 | 0.537 | 1.857 |

Table 8: Ablation study on the sensitivity of codebook refinement to the number of thinking traces used. We vary the number of traces used for instruction improvement ($N_{inst}$) and rubric improvement ($N_{rubric}$) on the Short Story (Engagement) task. All tested configurations yield better performance than the baseline.

**Effectiveness on Generalist Models**  To address concerns regarding the necessity of specialized reasoning models, we evaluated our framework on GPT-4o, a strong generalist model without native "thinking" output. Table 9 reports the SFT performance, showing that despite GPT-4o lacking native reasoning traces, our alignment method significantly improves its agreement with human raters (Average $\tau$ improves from 0.197 to 0.237). This trend holds for codebook refinement as well; as shown in Table 10, utilizing traces from GPT-4o yields consistently better evaluation guidelines than the original codebook (Average $\tau$ increasing from 0.316 to 0.339). These results confirm that the benefits of our framework extend to general-purpose LLMs.

| Task | Original Model | | | SFT (DeepSeek) | | | SFT (gpt-oss) | | | SFT (GPT-4o) | | |
|------|:---:|:---:|:---:|:---:|:---:|:---:|:---:|:---:|:---:|:---:|:---:|:---:|
| Metric | $\tau$ | ICC$_3$ | MSE | $\tau$ | ICC$_3$ | MSE | $\tau$ | ICC$_3$ | MSE | $\tau$ | ICC$_3$ | MSE |
| Story (C) | 0.177 | 0.231 | 2.016 | 0.463* | 0.604* | 0.838* | 0.460* | 0.581* | 0.944* | 0.326 | 0.457* | 1.354 |
| Story (E) | 0.275 | 0.289 | 1.939 | 0.048 | 0.364 | 1.364 | 0.179 | 0.371 | 1.831 | 0.118 | 0.421 | 1.910 |
| Essay | 0.202 | 0.233 | 1.947 | 0.322* | 0.267 | 1.149* | 0.240 | 0.244 | 1.387* | 0.259* | 0.294* | 1.198* |
| Summary (C) | 0.141 | 0.186 | 2.904 | 0.276 | 0.319 | 3.476 | 0.243 | 0.331 | 2.970 | 0.191 | 0.302 | 2.388 |
| Summary (F) | 0.275 | 0.378 | 1.938 | 0.319 | 0.402 | 1.160* | 0.306 | 0.498 | 1.200* | 0.341 | 0.389 | 1.627 |
| Translation | 0.171 | 0.240 | 2.778 | 0.260* | 0.348* | 2.492 | 0.256* | 0.314* | 2.439* | 0.219 | 0.277 | 2.404* |
| Chatbot (C) | 0.162 | 0.288 | 2.049 | 0.297* | 0.403* | 1.815* | 0.263* | 0.433* | 1.897 | 0.187 | 0.311 | 2.085 |
| Chatbot (H) | 0.176 | 0.274 | 1.969 | 0.264* | 0.422* | 1.723* | 0.295* | 0.466* | 1.575* | 0.252* | 0.430* | 1.701* |
| **Average** | **0.197** | **0.265** | **2.193** | **0.281** | **0.391** | **1.752** | **0.280** | **0.405** | **1.780** | **0.237** | **0.360** | **1.833** |

Table 9: Effect of Reasoning-Enhanced SFT on Human-LLM Rater Agreement (`DeepSeek-R1-Distill-Qwen3-8B`) including GPT-4o (non-reasoning LLM).

| Task | Original Codebook | | | Refined (DeepSeek) | | | Refined (gpt-oss) | | | Refined (GPT-4o) | | |
|---|---|---|---|---|---|---|---|---|---|---|---|---|
| Metric | $\tau$ | $ICC_3$ | MSE | $\tau$ | $ICC_3$ | MSE | $\tau$ | $ICC_3$ | MSE | $\tau$ | $ICC_3$ | MSE |
| Story (C) | 0.428 | 0.497 | 1.116 | 0.512 | 0.571 | 0.634* | 0.401 | 0.504 | 0.999 | 0.387 | 0.491 | 1.289 |
| Story (E) | 0.303 | 0.514 | 2.044 | 0.380 | 0.559 | 1.525 | 0.391* | 0.564 | 1.392 | 0.359 | 0.542 | 1.956 |
| Essay | 0.421 | 0.413 | 0.945 | 0.491* | 0.481 | 0.895 | 0.459* | 0.433 | 0.957 | 0.490* | 0.492* | 0.892 |
| Summary (C) | 0.166 | 0.150 | 2.861 | 0.188 | 0.178 | 2.591 | 0.177 | 0.181 | 2.668 | 0.169 | 0.159 | 2.904 |
| Summary (F) | 0.142 | 0.182 | 3.121 | 0.192 | 0.197 | 2.203 | 0.190 | 0.248 | 2.855 | 0.180 | 0.169 | 2.863 |
| Translation | 0.347 | 0.419 | 2.392 | 0.368 | 0.448* | 1.942* | 0.356 | 0.452 | 1.953* | 0.362 | 0.455* | 1.882* |
| Chatbot (C) | 0.354 | 0.451 | 1.920 | 0.370* | 0.485 | 1.724* | 0.378* | 0.484 | 1.804 | 0.379 | 0.481 | 1.798 |
| Chatbot (H) | 0.366 | 0.475 | 1.801 | 0.390* | 0.511* | 1.642* | 0.376 | 0.484 | 1.585* | 0.385* | 0.491 | 1.530* |
| **Average** | **0.316** | **0.388** | **2.025** | **0.361** | **0.429** | **1.645** | **0.341** | **0.419** | **1.777** | **0.339** | **0.410** | **1.889** |

Table 10: Effect of Codebook Refinement on Human-LLM Rater Agreement. Each value in the table is an average of the corresponding metrics calculated for each LLM rater.

## B  Dataset Details

**HANNA**  The HANNA dataset (Human-ANnotated Narratives for ASG evaluation)(Chhun et al., 2022) is a benchmark of 1056 stories generated from 96 prompts. Each story was annotated by 3 native English speakers from Amazon Mechanical Turk with Masters Qualifications along six defined evaluation criteria-engagement, complexity, surprise, relevance, coherence, and empathy, resulting in 19008 annotations. Among these, 10245 annotations exhibit a standard deviation lower than 1.0 across raters, which we regard as effective and reliable samples.

**Essay**  The Essay dataset consists of student essays from a Kaggle competition on automated essay scoring(Crossley et al., 2024). The 17307 student essays were annotated with a holistic quality score by expert raters. We regard all of them as effective samples.

**SummEval**  The SummEval dataset is a large-scale benchmark designed to evaluate summarization models(Fabbri et al., 2020). The summaries were generated by 23 summarization models trained on CNN/Daily-Mail news dataset. Each summary is evaluated by crowd-sourced annotators from Amazon Mechanical Turk and three expert annotators, where two had written academic papers on summarization for conferences, and one had completed a senior thesis on the topic. For our analysis, we focus on the expert annotations and retain only the effective samples, defined as those with a standard deviation below 1.0 across expert ratings. This filtering yields 5645 reliable samples out of 6400 total.

**WMT-2020**  The WMT20 Metrics Shared Task dataset is a benchmark for evaluating machine translation systems(Freitag et al., 2021). It contains system outputs across multiple language pairs (e.g., Chinese to English, English to German). Each translation was evaluated on both MQM (Multidimensional Quality Metrics) and SQM (Scalar Quality Metrics) by human annotators. In this work, we focus on the Chinese–English language pair and use the SQM (Scalar Quality Metric) annotations, where each translation is assigned three professional translators native in the target language of overall quality. The dataset contains 19,950 translation pairs, of which 6,934 exhibit a standard deviation below 1.0 across raters. We regard these as reliable samples and use them for training and evaluation.

**HelpSteer2**  The HelpSteer2 dataset is a preference dataset for training reward models that align large language models with human preferences(Wang et al., 2024). It contains 10,681 prompts, each paired with two responses, and every response is annotated on five dimensions: correctness, helpfulness, coherence, complexity, and verbosity. To ensure annotation reliability, each response is initially rated by three annotators, with two additional annotations collected when the disagreement among the first three annotations is high. In total, the dataset comprises 21,362 annotated responses, all of which we treat as effective samples. For our experiments, we focus on the dimensions of correctness and helpfulness.

## C    SFT Details

We trained all models using LoRA and chose AdamW as the optimizer. The hyperparameters are: learning rate = 5e-5 (warmup = 100 steps), batch size = 128, weight decay = 0.01, LoRA $\alpha$ = 32, LoRA r = 16.

## D    Prompts

---

**Instruction refinement**

\<instruction\>
You will be given a list of analysis on scoring the `{dimension}` of `{generation_type}`. Your task is to extract a concrete and concise step-by-step instruction from the analysis that could be easily followed by an annotator without any training. Based on your extraction, improve the original annotation guidelines (\<original_codebook\>...\</original_codebook\>). Only improve the instruction part (e.g. thinking path to follow), do not change the rubric part (e.g. criteria). Your final annotation guidelines should be put into \<codebook\>...\</codebook\> tag.
\</instruction\>

\<original_codebook\>
`{original_codebook}`
\</original_codebook\>

\<analysis\>
`{analysis}`
\</analysis\>

---

**Critique extraction from CoT**

\<instruction\>
You will be given an analysis on scoring the `{dimension}` of a `{generation_type}`. Your task is to list all the critique from the analysis that help evaluate the `{dimension}` of the `{generation_type}`. Your response should be a list of sentences, each on a new line, without bullet points. Your response should not include any other text.
\</instruction\>

\<analysis\>
`{cot}`
\</analysis\>

---

---

**Rubric refinement**

<instruction>
You will be shown a list of diverse critiques on the {dimension} of {generation_type}. Each of the critique is corresponding to a specific rating level. Your task is to summarize the critiques into a single rubric, based on the pattern of the different rating levels. Based on the new criteria, improve the original annotation guidelines. Only modify the criterion part. Your final annotation guidelines should be put into <codebook>...</codebook> tag.
</instruction>

<original_codebook>
{original_codebook}
</original_codebook>

<critiques>
{critiques}
</critiques>

---

## E  Complete Examples of Annotation Codebooks

A complete list of codebooks before and after refinement is provided in our anonymous codebase: `https://anonymous.4open.science/r/thru_judge_eye-56DD/codebooks/`. Here we show a complete refined version of Figure 4.

**Read the Materials Thoroughly**  Start by reading the prompt, the human story, and the subject story. Note that the subject story is the one being rated, not the human story. First, read the prompt to understand the context. Read the Human Story to get a sense of a complete narrative for comparison, if relevant. Read the Subject Story thoroughly, as this is the story you need to evaluate.

**Step-by-Step Rating:**

- Look for key elements such as characters, events, plot, themes, or setting . . .

- Evaluate if the identified elements are developed, interconnected, and coherent . . .

- Consider whether the story follows a structured progression (linear or non-linear) . . .

- Stories with minimal elements or undeveloped content are simpler. Stories that attempt multi-layered exploration of ideas, even if not fully realized, should be evaluated as more complex.

**Select the Rating**

1. The story is very simple, lacks depth, and introduces no intricate or developed elements.

2. The story is somewhat simple, with few elements that are underdeveloped, fragmented, or disconnected.

3. The story demonstrates some complexity, introducing multiple elements, but lacks depth, development, or coherence in integrating these elements.

4. The story is mostly complex, weaving together several developed elements with minor areas that could be expanded or refined.

5. The story is highly elaborate and complex, integrating multiple layers (e.g., detailed world-building, character depth, advanced structure, thematic sophistication) in a cohesive and interconnected way.

**Consider overall structure and presentation**

- Are there abrupt shifts, confusing elements, or fragmented storytelling that make it harder to engage with the story?

- Does the story have vivid details or emotional resonance that enhance engagement despite structural issues?

**Handle specific cases**

- If the story is cut off mid-sentence, rate it as if it ended just before the unfinished sentence.

- If the story is not relevant to the prompt, focus only on how engaging it is—irrelevance affects the Relevance criterion, not Engagement.

**Keep your focus**

- Rate solely based on the Engagement criterion, even if the story lacks relevance to the prompt.

- Do not let your assessment be influenced by the Human Story except as a potential comparison of narrative structure or clarity.

