# OpenReview forum: "Through the Judge's Eyes: Inferred Thinking Traces Improve Reliability of LLM Raters"
_TMLR — Accepted by TMLR_

### Review · Reviewer_okWo · 2025-11-29

**Summary Of Contributions:**

# Summary
The paper presents a framework that learns from inferred reasoning trace to improve the LLM as an automatic rater on downstream tasks. Key features include: 1) using rejection sampling to extract the chain of thought (CoT) tokens from models on LLM rating tasks; 2) the CoT can be seen as a proxy of the thought process of human raters when the predicted label agrees with human and thus can enhance LLM performance through SFT; 3) using the LLM summarize the inferred reasoning trace to rewrite the original code book (rating instruction and rubrics) to further improve the LLM rater performance.

# Strength
- The paper tackles an important task of using LLM as an automatic evaluator on open ended text generation tasks where the evaluation can be difficult and requires significant human efforts.
- The refinement of annotation code book is novel: it uses the inferred trace to rewrite the annotation code books and effectively improve the LLM performance. This method is also applicable to proprietary models which are black box LLM APIs.
- Extensive experiment results show the effectiveness of the proposed method.

# Weakness
- The method in Sec 3.1 is a standard method of distillation of reasoning via rejection sampling as mentioned in Section 2. The paper follows the existing method where LLM is used to sample a number of chain of thoughts and the ones that reach the correct final results are kept at the end for SFT. Therefore, the novelty of this method is weak.
- The codebook refinement process is empirical and there is no guarantee that the rewritten instruction and rubrics based on inferred thinking trace (not the actual human thinking) can improve the performance.
- Sec 3.2 Refining Annotation Codebooks: the choice of sampling 10 / 50 traces seems arbitrary.

**Audience:**

Yes

**Audience Explanation:**

The paper tackles the task of improving the LLM performance as an automatic rater of downstream tasks. The proposed method can effectively improve the alignment of LLM automatic rating with human rating. The proposed method is easy to adopt and is applicable to both open-weight and proprietary models. This work can potentially be useful for broader LLM research on open ended text generation tasks where the labeling requires significant human efforts.

**Claims And Evidence:**

Yes

**Claims Explanation:**

The proposed method is evaluated with experiments of SFT on thinking trace and codebook refinement using DeepSeek R1 and GPT OSS trace on various text generation tasks. The results show that both SFT on thinking trace and codebook refinement improve the LLM rater on metrics such as ICC_3, \tau, and MSE, which suggests better agreement with human rater and inter-rater reliability. Evaluation of codebook refinement for proprietary models such as Claude also suggests the proposed method is applicable to non-open-weights models.

**Requested Changes:**

See weakness.

---

> ### Author Response · Authors · 2025-12-25
>
> We thank the reviewer for your insightful feedback and for recognizing the significance of our work, the novelty of our codebook refinement approach, and the effectiveness of our results. We address your specific concerns below.
>
> ~[*Note: Per TMLR guidelines, we are waiting for the pending last review before uploading the PDF revision. We have implemented the changes locally and will upload the revised PDF once all reviews are posted.*]~ *The revision with requested changes is uploaded. Changes are colored in blue. Thank you for your patience!*
>
>
>
> > The paper follows the existing method where LLM is used to sample a number of chain of thoughts and the ones that reach the correct final results are kept at the end for SFT. Therefore, the novelty of this method is weak.
>
> Rejection sampling itself is indeed an established technique. The main contribution of our paper is not to advance this technique, but to leverage it to tackle the "important task of using LLM as an automatic evaluator on open ended text generation tasks where the evaluation can be difficult and requires significant human efforts" as precisely noted by the reviewer. The core methodological novelty lies in using these inferred traces to refine the annotation codebooks, which is precisely noted by the reviewer as novel and effective for both open-weight and proprietary models.
>
> > The codebook refinement process is empirical and there is no guarantee that the rewritten instruction and rubrics based on inferred thinking trace (not the actual human thinking) can improve the performance.
>
> Indeed,  there is no theoretical guarantee that inferred thinking traces can improve the rubrics.  Our approach is motivated by and aligned with the established practice in qualitative research to ask human raters to explain their decisions, discuss, and refine the codebook. We rely on rigorous empirical evaluation to validate the approach, focusing on two key metrics:
> 1. LLM-Human Alignment: This is the primary requirement for any automatic rater.
> 2. Inter-LLM Agreement (Inter-Rater Reliability - IRR): This demonstrates that the refined codebook generalizes across different LLM raters.
> As established in qualitative analysis literature, IRR is a core metric for developing effective codebooks [1]. Our results consistently show improvements in these metrics after codebook refinement. While a theoretical guarantee is beyond the scope of our study, the strong empirical evidence (ICC$_3$, $\tau$, and MSE) confirms the effectiveness of the method.
>
> > Sec 3.2 Refining Annotation Codebooks: the choice of sampling 10 / 50 traces seems arbitrary.
>
> To address this concern, we have done an additional sensitivity analysis (see table below). The different parameter configurations provide consistent improvement over the baseline, showing the effectiveness of the method is robust. While a comprehensive hyperparameter sweep could identify marginally optimal values, the core conclusions of the work regarding the method's effectiveness would remain unchanged. We have also added this result in Appendix A (Table 8).
>
> | #traces (instr / rubric) | $\tau$ | ICC$_3$ | MSE   |
> | :----------------------- | :----- | :------ | :---- |
> | Original (baseline)      | 0.303  | 0.514   | 2.044 |
> | 5 / 20                   | 0.370  | 0.548   | 1.622 |
> | 10 / 50                  | 0.380  | 0.559   | 1.525 |
> | 20 / 60                  | 0.396  | 0.549   | 1.737 |
> | 30 / 70                  | 0.363  | 0.537   | 1.857 |
>
> Reference:
> 1. MacQueen, Kathleen M., et al. "Codebook development for team-based qualitative analysis." Cam Journal 10.2 (1998): 31-36.

---

### Review · Reviewer_sFuU · 2025-12-01

**Summary Of Contributions:**

The paper looks at improving LLM-as-a-judge evaluation setups by utilising reasoning traces of reasoning-based LLMs for SFT. The authors investigate whether these silver-standard reasoning traces, used as a proxy for human reasoning traces, can improve LLM judges in their reliability. The authors present two setups - one is based on SFT using the reasoning traces for an improved judge, and the other method is using the reasoning traces for improving the annotation codebook. The authors show empirical improvements in inter-annotator agreement metrics for both these methods in this work. However, they do not conduct some important experiments that are crucial for clearly supporting their methods. They also do not provide enough description of related works, especially related to the use of reasoning traces in improving LLM performance across many tasks. I also believe the presentation of the paper may be improved, as there are many points where the Figures being referenced in the text are not close to where they are referenced.
I also do not believe the proposed method solves the issue at hand. The main reason for the use of LLM as a judge is the high cost of human evaluators. The paper looks at improving LLM evaluations, but requires significant human evaluation as a starting point for SFT to be effective. Further, an added cost of training a task-specific LLM evaluator is added to the LLM as a judge setup, making it less lucrative and difficult to reproduce for future researchers.

**Additional Comments:**

1. What happens in cases where none of the generated reasoning traces match with the human label? Further, could you provide an experiment analysing what happens if only those cases where the reasoning consistently matches with the human label, for example, atleast 50% (or any other choice) of the reasoning traces match with the human label.
2. The authors claim - For a model to reach the correct human label yi, its reasoning is more likely (but not guaranteed) to be a plausible approximation of the latent human thought process - it does not really need to be an approximation, for the tasks studied, it can be totally different to the interpretation of the human judge as well but still lead to the same outcome. This is evidenced by the fact that there are n = 0 to 16 valid reasoning traces that are produced by the LLM. Hence, how can each of them be an approximation if all of them are different?

**Audience:**

Yes

**Audience Explanation:**

LLMs are increasingly used as proxies for human evaluators on many machine learning tasks. Improvements to LLM as judge systems are of great interest to the ML research community.

**Claims And Evidence:**

No

**Claims Explanation:**

1. The authors present results on fine-tuning the base LM on the reasoning traces for improved reliability in the evaluations. However, it is not clear from these experiments whether the improved results are due to the inclusion of reasoning traces in the training setup or due to training itself. The authors need to disentangle this contribution of the SFT process, for example by training on just the human labels without the traces.
2. The various results tables show that there is not a statistically significant improvement in performance for close to half of the tested setups. This undermines the framework's application in untested domains.
3. The authors claim - "By providing more nuanced guidance, the refined codebook reduces this shared bias, causing the models to better reflect the natural and sometimes divergent distribution of human judgments". There is no evidence to support this claim. I hypothesise the opposite to be true as more detailed guidelines lead to an increased shared bias towards the aspects mentioned in the guidelines.
4. All results are shown on a single base model. It is not clear from these experiments if the setup is generalizable and can be applied to other choices of LLM judges.
5. The authors make many arbitrary choices (such as the selection of k = 16 generations, picking the 1st accepted reasoning trace, choice of a single 8B model for SFT). The reasoning behind these choices is not given, and it is important to understand the impact that different choices may have had.

**Requested Changes:**

1. Ablation on training using human labels only without reasoning traces - critical to securing your recommendation
2. Experiments on other choices of base LLM - critical to securing your recommendation
3. Evidence for the claim mentioned in point 3 above or a change in how it is framed- critical to securing your recommendation
4. experiments showing the impact of the choice of k = 16, how the reasoning trace is selected - strengthen the work

---

> ### Author Response · Authors · 2025-12-25
>
> We thank the reviewer for your constructive comments and for suggesting concrete experiments to strengthen our work. We have addressed your specific concerns and requested changes below.
>
> ---
> ### Responses to Requested Changes
>
> ~[*Per TMLR guidelines, we are waiting for the pending last review before uploading the PDF revision. We have implemented the changes locally and will upload the revised PDF once all reviews are posted.*]~
>
> *The revision with requested changes is uploaded. Changes are colored in blue. Thank you for your patience!*
>
> > Ablation on training using human labels only (without reasoning traces) - critical to securing your recommendation
>
> This is a good suggestion. We performed an ablation study (added to Section 4.5) where the base model was fine-tuned using only the human ratings (without reasoning traces) on the same dataset. We observed that the model trained only on ratings suffered an overall drop in performance compared to the trace-augmented model. This ablation confirms that the elicited intermediate thinking traces are an essential component to help the model ground its final judgment and align with human judgments.
>
> | Task | Original ($\tau$) | Original (ICC$_3$) | Original (MSE) | SFT w/ traces ($\tau$) | SFT w/ traces (ICC$_3$) | SFT w/ traces (MSE) | SFT w/o traces ($\tau$) | SFT w/o traces (ICC$_3$) | SFT w/o traces (MSE) |
> | :--- | :---: | :---: | :---: | :---: | :---: | :---: | :---: | :---: | :---: |
> | Short Story (Complexity) | 0.177 | 0.231 | 2.016 | 0.463* | 0.604* | 0.838* | 0.329 | 0.312 | 1.478 |
> | Short Story (Engagement) | 0.275 | 0.289 | 1.939 | 0.048 | 0.364 | 1.364 | 0.166 | 0.182 | 3.101 |
> | Student-Written Essay | 0.202 | 0.233 | 1.947 | 0.322* | 0.267 | 1.149* | 0.283* | 0.277* | 2.036 |
> | News Summary (Consistency) | 0.141 | 0.186 | 2.904 | 0.276 | 0.319 | 3.476 | 0.193 | 0.236 | 2.666 |
> | News Summary (Fluency) | 0.275 | 0.378 | 1.938 | 0.319 | 0.402 | 1.160* | 0.167 | 0.381 | 1.565 |
> | Translation | 0.171 | 0.240 | 2.778 | 0.260* | 0.348* | 2.492 | 0.217 | 0.272 | 2.763 |
> | Chatbot Response (Correctness) | 0.162 | 0.288 | 2.049 | 0.297* | 0.403* | 1.815* | 0.103 | 0.140 | 2.742 |
> | Chatbot Response (Helpfulness) | 0.176 | 0.274 | 1.969 | 0.264* | 0.422* | 1.723* | 0.072 | 0.120 | 2.430 |
> | **Average** | **0.197** | **0.265** | **2.193** | **0.281** | **0.391** | **1.752** | **0.191** | **0.240** | **2.348** |
>
> > Experiments on other choices of base LLM - critical to securing your recommendation
>
> To demonstrate generalizability, we replicated the SFT experiments using a different base model: DeepSeek-R1-Distill-Llama-8B (Added to Section 4.3 & Appendix). As shown in the table below, the method yields improvements similar to those observed with the original base model.
>
> | Task       | Original ($\tau$) | Original (ICC$_3$) | Original (MSE) | SFT ($\tau$) | SFT (ICC$_3$) | SFT (MSE) |
> | :----------------------------- | :---------------: | :----------------: | :------------: | :----------: | :-----------: | :-------: |
> | Short Story (Complexity)  |  0.324  |  0.419  |  1.552  | 0.315  |  0.419  | 1.248 |
> | Short Story (Engagement)  |  0.305  |  0.312  |  1.199  | 0.292  |  0.336  | 1.108 |
> | Student-Written Essay   |  0.075  |  0.087  |  4.475  | 0.237* | 0.260*  | 1.553* |
> | News Summary (Consistency)  |  0.059  |  0.0418  |  1.782  | 0.128  |  0.146  | 4.018 |
> | News Summary (Fluency)   |  0.159  |  0.203  |  1.792  | 0.213  |  0.359  | 1.203* |
> | Translation     |  0.174  |  0.143  |  2.145  | 0.236* |  0.173  | 2.028* |
> | Chatbot Response (Correctness) |  0.063  |  0.072  |  2.319  | 0.154* | 0.200*  | 2.046* |
> | Chatbot Response (Helpfulness) |  0.053  |  0.070  |  2.434  | 0.155* | 0.164*  | 2.228* |
> | **Average**     |  **0.151**  |  **0.168**  | **2.212** | **0.216** | **0.257** | **1.929** |
> > Evidence for the claim mentioned in point 3 above or a change in how it is framed- critical to securing your recommendation
>
> Thanks for pointing this out! As our main conclusion is that SFT using elicited reasoning traces improves the codebook and therefore improves the reliability of LLM raters, this statement is just one potential explanation of the mechanism. We agree that further analysis is needed to support this hypothetical explanation. We have reframed this claim in Section 4.4 to present it as a hypothetical explanation rather than a confirmed conclusion.
>
> > Experiments showing the impact of the choice of k = 16, how the reasoning trace is selected - strengthen the work
>
> The choice of k = 16 represents a pragmatic tradeoff between computational cost and data utilization rate (percentage of data examples that have at least one “valid” thinking trace that matches the ratings). We have added an analysis in Section 4.2: As illustrated in the plot, increasing k yields diminishing returns in data utilization rate. At k = 16, we already recover >90% of the examples that have at least one valid trace in k = 32 trials, but with half of the computational cost.

---

> ### Author Response · Authors · 2025-12-25
>
> ### Responses to Other Comments
>
> > … not provide enough description of related works, especially related to the use of reasoning traces in improving LLM performance across many tasks.
>
> We thank the reviewer for the suggestion to further elaborate on the related work. While reasoning traces have been used to improve various downstream tasks, existing work focuses on improving the performance within the scope of those specific tasks. Our work contributes a novel application: using reasoning traces to refine annotation codebooks, which we show improves the reliability of automatic raters, and it could be applied to many scenarios. We have expanded Section 2 to include a more comprehensive overview of related work in these related areas (including medicine, finance, cybersecurity, software engineering, and general-purpose distillation datasets).
>
> > The Figures being referenced in the text are not close to where they are referenced.
>
> We thank the reviewer for pointing this out. We have corrected the typesetting in the revised version to ensure figures appear near their textual references.
>
> > I also do not believe the proposed method solves the issue at hand... requires significant human evaluation as a starting point... added cost of training...
>
> We respectfully offer a different perspective regarding the cost-benefit analysis of our method. First, many of the subjective text generation tasks (such as the five applications we listed in the paper) are high-throughput and open-ended tasks, which require continuous evaluation of out-of-sample cases against human preferences. Our method maximizes the utilization of human effort in two ways: 1) we only require human ratings (which are fast and less costly to collect) rather than expensive human reasoning traces; 2) with a one-time fixed cost to collect initial human rating data and training the model, we obtain LLM judges that can be used to evaluate indefinite number of out-of-sample cases without additional human effort. Finally, while SFT introduces an initial overhead, it produces small, specialized LLM judges that are significantly cheaper to deploy and operate than prompting large proprietary models.
>
> > The various results tables show that there is not a statistically significant improvement in performance for close to half of the tested setups. This undermines the framework's application in untested domains.
>
> We thank the reviewer for pointing this out. We believe this is primarily due to the limited sample size of the test sets. We adhered to a common split strategy, using 10% of the dataset for validation and 10% of the dataset for testing. On those smaller subsets, it is difficult to show significance even when a considerable positive effect exists. Nevertheless, we find the directionality of the improvements is highly consistent. Our method outperforms the baseline in 7 out of 8 setups in the SFT scenario and all 8 out of 8 setups in the non-SFT scenario (for the primary metric $\tau$). This consistent surplus across diverse tasks suggests that the method is robust and generalizable.
>
> > (Follow up) What happens in cases where none of the generated reasoning traces match with the human label?
>
> If none of the 16 elicited reasoning traces match the human label, this data point will not be used for training. While this filtering criterion reduces the size of utilizable training data, it is a necessary step to prevent training the model on "hallucinated" reasoning traces. Future work could explore relaxed matching criteria to utilize more data for training.
>
> > picking the 1st accepted reasoning trace
>
> Picking the first valid trace is statistically equivalent to randomly selecting a valid trace because the generation order is stochastic. This "lazy" selection strategy is computationally efficient: once we find a match, we stop generating more traces.
>
> > How can each of them be an approximation if all of them are different?
>
> Humans have different thinking processes, even if they end up with the same conclusion. The diversity we observe in the elicited LLM's thinking traces mirrors this diversity of human reasoning. Therefore, the fact that multiple different traces lead to the same rating suggests the model has found multiple possible reasoning paths towards the human judgments. We have updated the text in Section 3.1 to frame this more precisely as generating "plausible justifications" that align with human judgments, rather than replicating a singular “ground-truth” thought process.

---

### Review · Reviewer_RgCH · 2026-01-02

**Summary Of Contributions:**

The authors try to improve the reliability of LLM judgments by analyzing the thinking processes of reasoning models. Although the authors performed simulations on different datasets, more simulations or clarifications are still needed.

__Strengths__

1. The authors employed various datasets, and also carried out some ablation studies.

2. The framework proposed in the article is intuitive and easy to understand.

__Weaknesses__

1. The framework in the paper is based on the rejection sampling method, but there is a lack of discussion regarding the rejection ratio.

2. The paper mainly used DeepSeek-R1 and gpt-oss-120b, and the authors could consider adding experimental results of non-reasoning LLMs as a comparison.

3. Temperature could also affect the efficiency and conclusions, especially since the authors used the same temperature parameter for different tasks.

**Audience:**

Yes

**Audience Explanation:**

As the authors mentioned, LLMs are now widely used in evaluation tasks, so it is important to improve their reliability.

**Broader Impact Concerns:**

This paper involves very few ethical concerns.

**Claims And Evidence:**

No

**Claims Explanation:**

The goal of this paper is to improve the results of RLMs, but how do they perform on these tasks compared to other LLMs? This has not been verified through experiments in the paper.

The idea of this paper is intuitive, but more experiments may be needed to demonstrate the generalizability of its conclusions, such as considering more parameters and prompts.

**Requested Changes:**

1. The authors need to add more experiments and explanations, referring to the weaknesses mentioned earlier.

2. The main results of the paper are presented in similar table formats, and the authors could consider using other forms to present and compare the results more effectively.

3. In addition to indicating cases of "statistically significant improvement over the baseline" in the tables, the authors could also highlight cases where performance has gotten worse.

4. It would be better to modify Figure 3, as the relationship between "Candidate Thinking Traces" and the left and right columns is unclear (for example, arrows could be added between the blocks).

---

> ### Author Response · Authors · 2026-01-12
>
> We thank the reviewer for their constructive comments and for suggesting concrete experiments to strengthen our work. We have addressed the specific concerns and requested changes below.
>
> > The framework in the paper is based on the rejection sampling method, but there is a lack of discussion regarding the rejection ratio.
>
> In the newly uploaded version, we have added an analysis of the data utilization rate (Figure 5) in Section 4.2. We define this rate as the percentage of data points for which at least one valid trace is found within $k$ generations; consequently, the rejection ratio is simply $1 - \text{data utilization rate at } k = 1$. Our results show that rejection rates are highly task-dependent, varying from 56% (Short Story) to 76% (Translation) in the single-shot setting. However, increasing $k$ effectively mitigates high rejection rates—for instance, data utilization for News Summary jumps from 36% at $k=1$ to 73% at $k=16$. This analysis confirms that while single-shot rejection can be high, our sampling strategy ($k=16$) ensures sufficient data coverage across diverse tasks while maintaining computational efficiency.
>
> > The paper mainly used DeepSeek-R1 and gpt-oss-120b, and the authors could consider adding experimental results of non-reasoning LLMs as a comparison.
>
> We appreciate the reviewer's suggestion to compare our approach against non-reasoning LLMs. We respectfully note that since our framework specifically relies on distilling explicit and detailed reasoning traces to refine codebooks and evaluators, a direct comparison with models that do not natively support such trace generation is somewhat beyond the scope of this work. Using Short Story (Complexity) as an example, the average thinking trace by DeepSeek-R1 is 340 words, while GPT-4o only generates 69 words, containing significantly less information. To address the reviewer’s interest in generalizability, we conducted an additional experiment using GPT-4o as the generator model. As shown in the updated results, our methods improve $\tau$ on average from 0.194 to 0.237 using SFT, and 0.316 to 0.339 using codebook refinement. Although less effective than using RLMs, this result confirms that our framework is not limited to specific reasoning models but can also benefit from non-reasoning LLMs. We have added this ablation study to Appendix A of the revised manuscript.
>
> *Effect of SFT on Human-LLM Rater Agreement (GPT-4o)*
>
> | Task | Original $\tau$ | Original ICC | Original MSE | SFT (GPT-4o) $\tau$ | SFT (GPT-4o) ICC | SFT (GPT-4o) MSE |
> | --- | --- | --- | --- | --- | --- | --- |
> | Story (C) | 0.177 | 0.231 | 2.016 | 0.326 | 0.457* | 1.354 |
> | Story (E) | 0.275 | 0.289 | 1.939 | 0.118 | 0.421 | 1.910 |
> | Essay | 0.202 | 0.233 | 1.947 | 0.259* | 0.294* | 1.198* |
> | Summary (C) | 0.141 | 0.186 | 2.904 | 0.191 | 0.302 | 2.388 |
> | Summary (F) | 0.275 | 0.378 | 1.938 | 0.341 | 0.389 | 1.627 |
> | Translation | 0.171 | 0.240 | 2.778 | 0.219 | 0.277 | 2.404* |
> | Chatbot (C) | 0.162 | 0.288 | 2.049 | 0.187 | 0.311 | 2.085 |
> | Chatbot (H) | 0.176 | 0.274 | 1.969 | 0.252* | 0.430* | 1.701* |
> | **Average** | **0.197** | **0.265** | **2.193** | **0.237** | **0.360** | **1.833** |
>
> *Effect of Codebook Refinement on Human-LLM Rater Agreement (GPT-4o)*
>
> | Task | Original $\tau$ | Original ICC | Original MSE | Refined (GPT-4o) $\tau$ | Refined (GPT-4o) ICC | Refined (GPT-4o) MSE |
> | --- | --- | --- | --- | --- | --- | --- |
> | Story (C) | 0.428 | 0.497 | 1.116 | 0.387 | 0.491 | 1.289 |
> | Story (E) | 0.303 | 0.514 | 2.044 | 0.359 | 0.542 | 1.956 |
> | Essay | 0.421 | 0.413 | 0.945 | 0.490* | 0.492* | 0.892 |
> | Summary (C) | 0.166 | 0.150 | 2.861 | 0.169 | 0.159 | 2.904 |
> | Summary (F) | 0.142 | 0.182 | 3.121 | 0.180 | 0.169 | 2.863 |
> | Translation | 0.347 | 0.419 | 2.392 | 0.362 | 0.455* | 1.882* |
> | Chatbot (C) | 0.354 | 0.451 | 1.920 | 0.379 | 0.481 | 1.798 |
> | Chatbot (H) | 0.366 | 0.475 | 1.801 | 0.385* | 0.491 | 1.530* |
> | **Average** | **0.316** | **0.388** | **2.025** | **0.339** | **0.410** | **1.889** |

---

> > ### Author Response · Authors · 2026-01-12
> >
> > > Temperature could also affect the efficiency and conclusions, especially since the authors used the same temperature parameter for different tasks.
> > > The goal of this paper is to improve the results of RLMs, but how do they perform on these tasks compared to other LLMs? This has not been verified through experiments in the paper.
> >
> > Our main conclusion is that the effective utilization of thinking traces inferred by RLMs improves LLM raters’ reliability, measured by their alignment with human judgment and agreement within different LLM raters. Our experiments demonstrate that both metrics improve when applying our method with a standardized, default temperature. While task-specific temperature tuning could potentially yield further performance gains, it would not change the validity of our findings. Therefore, we leave such optimization for future work.
> >
> > > The idea of this paper is intuitive, but more experiments may be needed to demonstrate the generalizability of its conclusions, such as considering more parameters and prompts.
> >
> > We have significantly expanded our experimental evaluation in the revised manuscript. Specifically, we added:
> > 1. **SFT Model Generalization:** Experiments using a different base model (DeepSeek-R1-Distill-Llama-8B) to verify that our findings hold across different architectures (Table 7);
> > 2. **Sensitivity Analysis:** An ablation study examining how the number of used thinking traces affects codebook refinement (Table 8); and
> > 3. **Component Validation:** An ablation SFT training on ratings only (without reasoning traces), which confirms the critical role of intermediate reasoning for model performance (Table 4).
> >
> > These additional results further improve the robustness of our conclusions across varying setups.
> >
> > > The main results of the paper are presented in similar table formats, and the authors could consider using other forms to present and compare the results more effectively.
> >
> > We appreciate this suggestion. To address it, we have added two figures in the revised manuscript:
> > 1. Trade-off between Data Utilization Rate and Number of Samples (Figure 5); and
> > 2. Effect of Codebook Refinement on Rater Agreement among LLM Raters (Figure 6).
> >
> > > In addition to indicating cases of "statistically significant improvement over the baseline" in the tables, the authors could also highlight cases where performance has gotten worse.
> >
> > Thank you for this suggestion. We have explicitly discussed the specific cases where performance decreased in Sections 4.3 and 4.4.
> >
> > > It would be better to modify Figure 3, as the relationship between "Candidate Thinking Traces" and the left and right columns is unclear (for example, arrows could be added between the blocks).
> >
> > We appreciate the reviewer’s suggestion. We have improved the clarity of this figure, including adding arrows between the blocks.

---

### Author Response · Authors · 2026-01-12

We thank all reviewers for their insightful and constructive comments on our work. We are glad to see that our proposed framework is considered “intuitive and easy to understand” by **Reviewer RgCH**. We are also encouraged that the codebook refinement method is recognized as “novel” and applicable to both open-weight and proprietary models by **Reviewer okWo**, who further noted that our “extensive experiment results show the effectiveness” of the method. Lastly, the importance of the problem we tackle—improving LLM reliability for evaluation tasks—was highlighted as being of “great interest” to the community by **Reviewers RgCH, sFuU, and okWo**.

We have responded to individual questions below. We have also made several modifications to our paper to include additional experiments (e.g., SFT baseline without reasoning, data utilization analysis, and sensitivity tests) and clarify our contributions, highlighted with the blue font.

---

### Decision · Action_Editor_qcVd · 2026-02-03

**Recommendation:** Accept as is

**Audience:**

Yes

**Audience Explanation:**

LLM as a judge is a very popular technique, but a common way of improving performance requires collecting human thought traces, which is expensive. The authors propose a way to sidestep that so this may be of interest to the community.

**Claims And Evidence:**

Yes

**Claims Explanation:**

The evidence seems accurate and clear, one of the reviewers has some slight concerns remaining about how convincing it is (due to experiments being performed on relatively few LLMs) but generally it is above the threshold.